# Exploring the perceived impact of physical activity on physical and mental health among individuals with long COVID: A qualitative interview inquiry

Zoe Sirotiak[1]*, Ann M. Oberhauser[2], Arie M. Sirotiak[3], Kate A. Nettleton[4], Duck-chul Lee[5], Emily B. K. Thomas[6], Angelique G. Brellenthin[1]

1 Department of Kinesiology, Iowa State University, Ames, Iowa, United States of America, 2 Department of Sociology and Criminal Justice, Iowa State University, Ames, Iowa, United States of America, 3 Department of English, Iowa State University, Ames, Iowa, United States of America, 4 Department of Psychological Sciences and Counseling, Youngstown State University, Youngstown, Ohio, United States of America, 5 Department of Health and Human Development, University of Pittsburgh, Pittsburgh, Pennsylvania, United States of America, 6 Department of Psychological and Brain Sciences, University of Iowa, Iowa City, Iowa, United States of America

* zmsiro@iastate.edu

## Abstract

### Objectives

Long COVID presents a significant health burden with limited treatment options. Physical activity (PA) has been suggested for self-management, yet symptom worsening has been reported in similar patient populations. This study aims to identify PA's perceived impact on physical and mental health in adults with long COVID.

### Methods

Semi-structured interviews were conducted with 34 adults (mean age 52 years, 62% women) based in the United States (U.S.) self-reporting long COVID. PA-related content was analyzed using deductive thematic analysis, assessing worsened and improved health experiences attributed to PA.

### Results

Participants' perceived health scores were one standard deviation worse than the general U.S. population. Most participants (64.7%) reported worsening long COVID symptoms with PA, while 14.7% noted improvement. Themes for worsened physical health included post-exertional malaise, specific symptom worsening (e.g., fatigue), limited PA abilities, external control perceptions, forced inactivity, and loss of previous PA abilities. Improved health themes involved beliefs in health benefits, symptom improvement, increased energy, accomplishment, enhanced PA abilities, and hope.

**Data availability statement:** The data supporting the findings of this study include an interview guide and codebook. These materials are publicly available on the Open Science Framework at https://doi.org/10.17605/OSF.IO/6ZW4H. Full interview transcripts are not shared due to ethical and confidentiality considerations as approved by the Iowa State University Institutional Review Board. The Iowa State University Institutional Review Board may be contacted at irb@iastate.edu or 515-294-4566.

**Funding:** The author(s) received no specific funding for this work.

**Competing interests:** The authors have declared that no competing interests exist.

## Discussion

PA's impact on health varied among individuals with long COVID, highlighting the need to tailor PA recommendations to individual needs and limits.

## Introduction

Long COVID, or the experience of new or worsened chronic symptoms following SARS-CoV-2 infection, affects between 10−36% of adults following SARS-CoV-2 infection [1–4]. Individuals with long COVID often report worsened physical health, mental health, and functional abilities [5–9]. Reported duration of long COVID varies considerably, with some suggesting significant improvement in months, while others continue to experience symptoms years following infection [10–13]. Given the lack of clear cause and variation in symptom burden and duration, evidence-based treatment and management recommendations for long COVID are sparse [4,14].

Physical activity (PA) is a common treatment strategy in the context of somatic syndromes, a group of conditions characterized by unexplained physical symptoms [15,16]. Long COVID often presents comorbidly with these conditions [17], and PA has also been proposed as a potential self-management strategy for long COVID [18]. However, recommendations encouraging PA, particularly exercise therapy programs, have been criticized by some researchers and patient advocacy groups in the somatic syndrome community [19–22]. Concerns regarding PA recommendations include poor methodological quality of studies supporting exercise, harmful effects of post-exertional malaise, lack of consideration of lived experience, and dismissal of physiological symptoms as psychological in origin [19–22].

The effect of PA on long COVID symptoms is unclear, though preliminary work has advocated caution. Those with long COVID have been reported to be less active and require more physical assistance compared to their pre-COVID status [23,24]. Many with long COVID report worsening of symptoms with PA, while relatively few reported improvements following a bout of PA [23–26]. However, most studies have focused specifically on the impact of PA on long COVID symptoms, without identifying the effects of PA on perceived physical and mental health or lived experience.

Health perception, or an individual's beliefs regarding their health status, indicates the degree to which an individual feels they have the capacity to function in physical, mental, and social domains [27,28]. While specific symptoms are frequently assessed in long COVID-related investigations, perceived health allows for an evaluation of how these symptoms impact an individual's function in their personal context. Instead of considering biometric data such as blood pressure or heart rate, perceived health allows consideration of lived experience. Given the lack of treatment options available to those with long COVID, determining the impact of PA, a suggested self-management strategy, is vital as millions continue to be affected. The purpose of this study is to investigate the perceived impact of PA on physical and mental health among individuals with long COVID. We aim to contextualize the perceived effect of PA on physical and mental health as relayed through semi-structured in-depth interviews conducted among individuals with self-identified long COVID.

## Materials and methods

### Ethics statement

The project was approved by the Iowa State University Institutional Review Board (IRB; IRB ID: 23–158). The Iowa State University IRB did not require formal written or verbal consent as this project was deemed exempt under Exemption 2 (surveys, interviews, educational tests, observation) of the 2018 Common Rule. This research project aligned with Exemption 2 in that it involved only survey and interview procedures, and the IRB conducted a "limited IRB review," determining that adequate privacy and confidentiality practices were in place. Participants read through a consent information sheet that contained complete study information and if they chose to participate, participants indicated their clear informed consent by clicking forward on the screen. Participants also gave verbal consent for recording of their interview prior to beginning the interview. Participants had the option to skip any question that they did not wish to answer during both the survey and the interview. Potentially identifying information in the transcripts was generalized prior to coding.

### Participants and data collection

Semi-structured in-depth interviews were conducted virtually among a sample of individuals with self-reported long COVID. Participants were recruited from a preceding survey study investigating the experiences of individuals with long COVID [6–8]. Individuals in the original study were recruited through mass email and long COVID-specific social media sites. Those self-endorsing long COVID who completed the original survey (n = 379) had the option to offer their names and contact information for future studies. These individuals were recruited for interviews, with others snowball sampled from these initial participants. Inclusion criteria included being ≥18 years old, based in the United States, the ability to communicate verbally in English, and reliable internet and telephone service and devices. The recruitment period occurred from May 15, 2023 to August 1, 2023.

Prior to the interview, participants completed a brief accompanying survey, detailing sociodemographic characteristics, long COVID symptoms, physical and mental health status, and PA behavior and experiences. An interview guide was prepared to probe specific aspects of the lived experience, including physical and mental health, changes in PA, adjustments to plans, goals, and beliefs since developing long COVID, as well as the social experience of long COVID. The interview guide was developed by a physical therapist (ZS), with input from a qualitative methods expert (AO), a PA and health expert (AB), and a clinical psychologist (ET). Due to the semi-structured nature of the interview and limited available participants, pilot testing was not conducted, with interviews being flexible in response to participant experiences shared. Experiences with PA was one focus of the larger interview, which comprehensively assessed experiences with long COVID. The interview guide and additional information about the interview study is available on OSF [29].

### Measures

**Sociodemographics.** Sociodemographic characteristics included age, gender, race, ethnicity, education, and income.

**Physical activity level.** Participants selected whether they typically engaged in 0 minutes, 1 to <150 minutes, 150 minutes to <300, or ≥300 minutes or more of moderate or vigorous aerobic physical activity per week. These response options were selected according to established aerobic physical activity categories based on the 2018 Physical Activity Guidelines for Americans (0 min = inactive; 1 to <150 min = insufficiently active; 150 to <300 min = active; ≥300 min = highly active) [30].

**Change in exercise level.** Participants selected whether they currently perform less exercise, about the same amount of exercise, or more exercise than they used to before spring 2020.

**Perceived impact of PA.** Participants selected how they perceived PA or exercise to affect their long COVID symptoms: makes worse (worsened), does not affect (unchanged), and makes better (improved).

**Perceived physical and mental health.** Physical and mental health were measured with the Patient Reported Outcomes Measurement Information System (PROMIS) Global Health, version 1.2 [31]. The measure considers quality of life, satisfaction with social activities and roles, functional abilities, fatigue, and pain [32]. Physical and mental health subscales can be calculated, with four items on each subscale. Raw scores can be converted to T-scores based on U.S. adult normative data, with a mean of 50 and a standard deviation of 10 [33]. Higher scores indicate better physical or mental health. Internal consistency in the current sample was acceptable (physical health α = 0.77; mental health α = 0.81).

## Qualitative analysis

Analyses involved three coders (ZS, AS, KN). One coder (ZS) was involved in project conceptualization and data collection, while the other coders (AS, KN) were involved in coding and analysis only. A deductive thematic analysis method was used to code the interview transcripts, in which preconceived categories based on existing knowledge and research questions provided a structure for coding [34–36]. Transcripts were coded according to the categories shown in Table 1 to explore the varied positive and negative physical and mental health effects of PA noted among participants with long COVID. Codes were generated within each category as they arose in transcripts [35,36].

First, coders individually read through the transcripts from each interview and developed a list of potential codes in each impact of PA category [36,37]. The coders then met to establish a preliminary codebook. Following establishment of the codebook, the coders individually coded each transcript, assigning 0 (not present) or 1 (present) to each preliminary code for each participant. Codes with at least two coders endorsing them were considered endorsed, while codes with one coder endorsing were discussed by the group of coders until agreement was reached. The final number of participants endorsing each code was noted, and overall themes emerging from the codes was determined in conversation among the coders.

## Results

Thirty-four adults in the United States reporting long COVID participated in the in-depth semi-structured interviews. Table 2 lists demographic information for the interviewees. Participants were an average age of 51.6 (SD = 17.0) and most identified as female (61.8%), white (97.1%), and not Hispanic or Latino (97.1%). Most of the participants had at least a bachelor's degree (79.4%). The sample reported perceived physical and mental health approximately one standard deviation worse than that of the U.S. adult population, indicating notably poorer health than the general U.S. adult population [33]. Most participants were either inactive (38.2%) or insufficiently active (38.2%), and almost all (94.1%) reported engaging in less PA than before spring 2020. Most of the participants noted that PA seemed to make their long COVID-related symptoms worse (64.7%), while 14.7% reported improvement in long COVID-related symptoms with PA.

The physical activities reported by the participants can be viewed in Table 3. Among the most common PA types endorsed by the participants were walking for exercise (55.9%), walking a dog (23.5%), activities of daily living (20.6%), and weightlifting (17.6%). The interviews were coded into worsened and improved perceived physical and mental health categories, with resulting codes presented in Table 4 and descriptions of codes in S1 Table.

**Table 1. Deductive thematic analysis coding categories.**

| Impact of PA on… | Worsen | Improve | No change |
|---|---|---|---|
| Physical health | | | |
| Mental health | | | |

a PA = physical activity.

**Table 2. Interview participant characteristics.**

| | Total (n = 34) |
|---|---|
| Age M years (SD) | 51.6 (17.0) |
| Gender identity, N (%) | |
| Male | 12 (35.3) |
| Female | 21 (61.8) |
| Genderqueer/gender non-conforming | 1 (2.9) |
| Race, N (%) | |
| White | 33 (97.1) |
| Not white | 1 (2.9) |
| Ethnicity, N (%) | |
| Hispanic or Latino | 1 (2.9) |
| Not Hispanic or Latino | 33 (97.1) |
| Education, N (%) | |
| High school | 4 (11.8) |
| Associate/technical degree | 3 (8.8) |
| Bachelor's degree | 14 (41.2) |
| Graduate school/professional | 13 (38.2) |
| Income, N (%) | |
| Less than $16,000 | 2 (5.9) |
| $16,000 - $34,999 | 2 (5.9) |
| $35,000 - $49,999 | 3 (8.8) |
| $50,000 - $74,999 | 5 (14.7) |
| $75,000 – $99,999 | 2 (5.9) |
| Greater than $100,000 | 16 (47.1) |
| Unsure | 4 (11.8) |
| Health, M T-score (SD) | |
| Physical health | 38.29 (7.66) |
| Mental health | 39.56 (8.67) |
| Physical Activity, N (%) | |
| Inactive | 13 (38.2) |
| Insufficiently active | 13 (38.2) |
| Active | 7 (20.6) |
| Highly active | 1 (2.9) |
| Change in physical activity, N (%) | |
| Less | 32 (94.1) |
| Same | 1 (2.9) |
| More | 1 (2.9) |
| Perceived change in health with PA | |
| Makes worse | 22 (64.7) |
| Does not affect | 7 (20.6) |
| Makes better | 5 (14.7) |

[a] M = mean; SD = standard deviation; N = number.

**Table 3. Types of PA reported by interviewees.**

| Type of PA | Number reporting (%) |
|---|---|
| Walking- unspecified | 19 (55.9) |
| Walking dog | 8 (23.5) |
| Activities of daily living | 7 (20.6) |
| Weightlifting | 6 (17.6) |
| Physical therapy exercises | 5 (14.7) |
| Yoga and/or stretching | 5 (14.7) |
| Cycling (indoor or outdoor) | 5 (14.7) |
| Active job | 5 (14.7) |
| Yardwork | 4 (11.8) |
| Hiking | 3 (8.8) |
| Swimming or pool exercises | 3 (8.8) |
| Stairs | 2 (5.9) |
| No activity | 2 (5.9) |
| Running | 1 (2.9) |
| Bowling | 1 (2.9) |
| Fishing | 1 (2.9) |
| Dancing | 1 (2.9) |

[a] PA = physical activity.

Themes derived from the data are listed in Table 5. Three themes emerged detailing worsened physical health with PA: 1) post-exertional malaise; 2) worsened specific symptoms; and 3) symptoms as a limiting factor in performing PA. Similarly, three themes emerged from codes describing improved physical health with PA: 1) believing that exercise is important for health; 2) improved symptoms associated with PA; and 3) improved energy. Worsened mental health themes included 1) perceived external control of PA; 2) living a forced inactive lifestyle; and 3) loss of former abilities. Three improved mental health themes emerging included 1) sense of accomplishment; 2) improvements in PA abilities; and 3) increased hope.

## Discussion

The proportion of participants noting negative effects of PA on physical health was substantial, with the negative effects of PA represented in the three most frequently endorsed codes. This result seems to fit with the higher percentage of interview participants reporting worsened health (64.7%) compared to improved health with PA (14.7%). These results also align with other reports of PA associated with worsening health among individuals with long COVID [23,25,26]. Post-exertional malaise, associated with PA in other qualitative work in the context of long COVID [25,26] was also a theme emerging from our codes, with 64.3% of participants noting feeling worse following PA and 41.2% reporting worsened symptoms the day following PA. The loss of former abilities theme developed from our data, accompanied by grief associated with changes in PA, fit with the sense of loss associated with PA changes noted in other qualitative long COVID work [25].

The current study assessed moderate or vigorous aerobic PA, and the 64.7% of participants with long COVID reporting worsening of symptoms with PA in the current sample is similar to the proportions noting worsening of symptoms with both moderate and vigorous intensity exercise in another long COVID sample [23]. While prior work has noted 0.84% of a sample with long COVID experienced improved symptoms with PA [23], the percentage of individuals reporting improvement with PA was notably higher in the present study (14.7%). The source of this discrepancy is unclear, but the participants in the present study had the ability to complete an in-depth interview while the comparison study utilized a survey [23], and

**Table 4. Impact of PA on perceived health: Interview results.**

| Worsened | N (%) | Themes | Improved | N (%) | Themes | Unchanged | N (%) | Themes |
|---|---|---|---|---|---|---|---|---|
| **Physical Health** | | | | | | | | |
| Fatigue/exhaustion | 27 (73.5) | 1,2 | "Exercise is good" belief | 11 (32.4) | 4 | Unchanged by PA | 3 (8.8) | 13 |
| Crash/feel worse after | 25 (64.3) | 1 | Feels better | 7 (20.6) | 5,6 | | | |
| Worse next day | 14 (41.2) | 1 | Improved in nature | 5 (14.7) | 5,6 | | | |
| No energy for PA | 11 (32.4) | 3 | Energized by exercise | 4 (11.8) | 6 | | | |
| Heart rate (too slow or fast) | 10 (29.4) | 2 | | | | | | |
| Shortness of breath | 8 (23.5) | 2 | | | | | | |
| Can't get out of bed | 8 (23.5) | 1,3 | | | | | | |
| Must take breaks | 8 (23.5) | 3 | | | | | | |
| Dizziness/balance issues | 6 (17.6) | 2 | | | | | | |
| Impaired thermoregulation | 3 (8.8) | 2 | | | | | | |
| Chest pain | 3 (8.8) | 2 | | | | | | |
| Worsened other conditions | 2 (5.9) | 2 | | | | | | |
| Undesired changes in weight | 2 (5.9) | 2 | | | | | | |
| **Mental Health** | | | | | | | | |
| Cannot do what they used to | 26 (76.5) | 7,9 | Mood improved after | 11 (32.4) | 10,11 | Unchanged by PA | 4 (11.8) | 14 |
| Forced inactive lifestyle | 25 (73.5) | 8 | Sense of accomplishment | 11 (32.4) | 10 | | | |
| Frustration | 23 (67.6) | 7 | Feeling hopeful | 9 (26.5) | 12 | | | |
| Cannot do what they want | 23 (67.6) | 7,8 | Noting improvements | 6 (17.6) | 11 | | | |
| Grief or loss | 18 (52.9) | 9 | Being outdoors | 5 (14.7) | 11,12 | | | |
| Unpredictability of PA effects | 15 (44.1) | 7 | | | | | | |
| Fear/anxiety regarding PA | 11 (32.4) | 7 | | | | | | |
| Sadness | 11 (32.4) | 8,9 | | | | | | |
| Anger | 6 (17.6) | 7,8 | | | | | | |
| Trapped | 5 (14.7) | 7 | | | | | | |
| Sensory overload | 5 (14.7) | 8,9 | | | | | | |
| Regret following PA | 4 (11.8) | 7,9 | | | | | | |

[a] "Themes" indicate codes driving themes as detailed in Table 5.

**Table 5. Perceived effect of PA themes from interviews.**

| Impact of PA on… | Worsened | Improved | Unchanged |
|---|---|---|---|
| Physical health | -Post-exertional malaise[1]<br>-Specific symptoms[2]<br>-Symptoms as a limiting factor[3] | -Exercise is compulsory to health[4]<br>-Improved symptoms[5]<br>-Improved energy[6] | -No change with PA[13] |
| Mental health | -External control of PA[7]<br>-Forced inactive lifestyle[8]<br>-Loss of former abilities[9] | -Sense of accomplishment[10]<br>-Improvements noted[11]<br>-Hope increased[12] | -No change with PA[14] |

[a] Superscript numbers indicate themes derived from codes in Table 4.

individuals who able to complete an interview may be less affected than others. The percentage of participants noting no effect of PA on long COVID symptoms in the present study (20.6%) was lower than the 28.7% reported by Wright et al. (2022). Differences in sample size and consideration of physical activity intensities by Wright et al. (2022) compared

to non-specific physical activity in the current study may contribute to these differences noted. Overall, it seems that the variation in perceived effects of PA on health in the current sample generally aligns with published literature.

The most frequently noted effects of PA on physical health were fatigue or exhaustion during and/or following PA (73.5%), similar to other work noting fatigue as the most common worsened symptom associated with PA in a long COVID sample (68.4%) [23]. Crashing or feeling worse after PA (64.3%) was the next most frequently endorsed effect of PA, and along with worsening next day (41.2%), this code drove the post-exertional malaise theme, fitting with literature reporting post-exertional malaise as a primary symptom of long COVID [38]. "I'm usually shot the whole next day. Like, my symptoms get worse… I'm too tired to do anything," one interviewee reported of their attempts to return to an exercise routine. "If I do something one day, I can pretty much plan that I'm going to be in bed most of the day for the whole next day or the day after," another noted.

Worsening of specific long COVID symptoms was a second theme that emerged, highlighting the variety of symptoms reported with PA in the sample. Fatigue or exhaustion was the most endorsed symptom (73.5%), fitting with extensive literature reporting fatigue as the most prevalent symptom of long COVID [39,40]. Undesirable heart rate changes, including tachycardia, were noted among 29.4% of the participants. While it is unclear if the participants would meet diagnostic criteria, postural orthostatic tachycardia syndrome has been a documented comorbidity of long COVID [41,42], and exercise, particularly in an upright position, has been noted as a trigger among individuals with this condition [43]. Shortness of breath, dizziness or balance issues, impaired thermoregulation, and chest pain, each long COVID symptoms [44,45], were among other symptoms noted by the sample to be precipitated by PA.

The impact of experience on determining PA level was a third theme, with prior PA experiences limiting current PA. "If I … run, it'll just make everything worse, rather than making it better. From experience, I've tried running. It didn't work," one participant noted. While some expressed a desire to return to their prior activity level, many found this difficult, fitting with other reports of attempts to return to PA with long COVID [24,26]. One participant noted an attempt at returning to their prior level of activity, with deleterious effects: "I've ended up regretting it either that day or the next day." Some participants noted a slow activity decline as they continually experienced worsened symptoms with prior PA routines, resulting in their current inactive state. These declines fit with other literature reporting steep drops in the self-reported proportion of individuals meeting PA guidelines before (83.7%) and after (8.2%) long COVID [23]. A history of worsened health with PA limited attempts to be active among some, with participants noting that while low PA levels may be detrimental to their health, the worsened symptoms associated with PA seem more so. Several had given up trying to return to their prior PA level given multiple unsuccessful attempts, with one participant reporting: "I don't even bother trying to do anything because I know what the outcome is." Others reported a change in how they view PA: "I mean, I definitely view it as having a more negative effect on my life. I (was not) like really a huge fan before (long COVID). But I knew objectively it was a positive thing. Now I wouldn't say that I think of it that way."

Other participants, however, noted improvements in physical health with PA. Nearly a third (32.4%) of participants reported that they knew PA was good for their health, motivating them to attempt it. These participants reported believing that increasing PA is likely to help their symptoms as they have been educated that PA has positive impacts on health. This finding is consistent with other literature demonstrating that knowledge of positive effects of PA is associated with higher PA levels [46]. Some participants endorsing this belief noted that they believed that their symptoms were temporary, and that PA would assist in recovery from long COVID. Other participants noted improvements in both long COVID symptom burden and PA abilities, which some linked together. PA was a mechanism through which others could measure their recovery process, with one participant noting: "I've been able to go hiking (recently)… I can go two and a half miles, yay! So that's amazing. I could not have even considered that (a few months ago)." Participants noting improvements in long COVID symptoms drove the emerging improved symptoms theme. While several participants noted post-exertional malaise, others noted that PA energized them, allowing them to be more productive for the rest of the day. Participants reporting improvements in energy often connected PA with nature, such as enjoying fresh air while walking outdoors.

Several noted that their long COVID symptoms seemed to improve in nature, and PA allowed an opportunity to exercise and interact with nature again after being ill, increasing their energy to perform other tasks. The positive role of nature noted among these individuals coincides with other work associating high PA level and nature exposure with positive physical health effects [47].

Similar to physical health, negative effects of PA on mental health were endorsed by a significant proportion of participants, as demonstrated in Table 4. Several noted a feeling of external control of PA, with little personal autonomy: "Well it's my nemesis. I know that I can't do it. Of course, I would love to do it, but I know I can't do it…. I know what it does to me. So, it's not that I wouldn't want to do it. Of course, I would want to do it, but I know what it does." Others noted frustration with healthcare providers and treatment options presented, noting a significant lack of education regarding appropriate PA strategies for long COVID: "They felt I should be doing physical therapy… so I went to this guy and… he had no idea what he was doing and almost immediately put me into a panic attack and that was the end of it." Another noted: "(The healthcare provider) doesn't know what he's doing. (They said) you get better by, you know, working up to it, working up to exercise. You don't. It just makes things worse." Many expressed frustrations about the lack of established PA options for those with long COVID: "(The clinic) said they were finding that physical therapy is effective. But they admitted it's because they didn't really have any other good ideas. (And it's) only effective if it is customized (to long COVID)." Some noted that while healthcare providers suggested that improving long COVID was within their control such as through increasing PA, they found this to be counterproductive as their symptoms worsened with PA. The frustration with healthcare providers seemed to relate to other literature noting conflicting PA recommendations from healthcare providers among individuals with long COVID [23,24]. Participants often noted lack of long COVID education among healthcare providers, eroding the participant's trust in medical recommendations, including recommendations regarding PA. Others noted that the low amounts of PA that they currently engage in are not due to a lack of desire, but instead because of lived experience of attempting PA. These concerns drove the external control of PA theme, with feelings of little control of activity level.

Self-determination theory, which posits that humans have three innate psychological needs [48], may be a framework through which the external control of PA noted may be understood. These three proposed needs of self-determination theory include competence, autonomy, and relatedness, with motivation and quality of life significantly impacted if individuals are unable to meet them [48,49]. Individuals with long COVID have noted loss of autonomy related to symptoms, resulting in wide-ranging impacts across social and professional roles [50]. PA experiences among participants in the present study seemed to align with published literature, suggesting that autonomy may be limited by poor personal experiences with PA, distrust of healthcare provider PA recommendations, and feelings of uncertainty regarding the role of PA in long COVID management. Self-determination theory argues that individuals with feelings of limited autonomy may experience psychological distress, which was present among several individuals in the present study, with some noting that they desire to engage in PA but feel unable to do so safely or effectively. The feelings of loss of competence regarding PA also aligns with the impaired feelings of competence noted by self-determination theory to contribute to psychological distress. Individuals with long COVID often report notable loss of socioemotional and occupational roles [51], indicating that perceptions of competence may be substantially affected beyond PA to other roles and commitments.

Other participants noted a forced inactive lifestyle, often noting that they used to live active and busy lives before long COVID: "It's very disheartening to sit and watch your family do the things you want to do and not be able to participate. I was like, okay, is this it? Is this all I'm going to be able to do? That's very, very frustrating." Some participants noted that PA was an important part of their lifestyle and identity prior to long COVID, and they have had to develop a new, less active lifestyle due to their symptoms, a finding mirrored in the long COVID literature [24,26,52]. Fear of PA and uncertain consequences were noted among other participants, causing them to avoid PA: "Well I was kind of afraid to exercise because it seemed like intuitively that if I was so fatigued, maybe I shouldn't be exercising." These concerns sometimes drove lower activity levels than desired by the participant.

The loss of former abilities was another theme that emerged from the interviews. As detailed in prior literature [24,26], participants with PA-related identities prior to long COVID felt particularly affected: "(PA) was a really good stress reliever. It made me feel good about myself. And a lot of things like that are gone now. Sometimes I think to myself, who the hell is this guy?" Another participant noted initial hope after recovery from acute SARS-CoV-2 infection quickly turning to disappointment as the loss of athletic ability became clear: "And it's more depressing sometimes to try… it's just hard when you think you can (return to activity) and you get out there and you get excited and it's just like no, no, no." Others noted loss of an identity with their new physical status, with one saying of their current self: "I don't know (who) that person is."

Other participants noted positive effects of PA on their mental health. Some pointed to the sense of accomplishment associated with PA, particularly after being limited due to long COVID: "(PA) impacts my mood a lot. It makes me happy for the rest of the day if I can get out and do something." For others, PA was a method of observing their progress in recovering from long COVID, lifting their mental health. Particularly among individuals who reported being active prior to long COVID onset, returning to prior activities was a tangible sign that their long COVID symptoms were subsiding, or that their ability to manage long COVID symptoms was improving.

These codes drove the improvements noted theme, in which both improvements in long COVID symptoms were tied to PA abilities. Particularly among individuals who noted reduction in long COVID symptoms, hope for further improvement in both symptoms and a return to an active lifestyle was another theme that emerged. These participants were encouraged by their PA abilities, and some noted that improved PA abilities translated into improved functional abilities in their daily lives. Still, this hope was sometimes accompanied by wariness, highlighting participants who noted both positive and negative impacts of PA, sometimes unsure how PA will affect them until they attempt it. One interviewee commented that while PA can bring health benefits and hope for recovery, the possibility of pushing too far remains: "Well there's certainly this catch-22. So, like, going outside and seeing sunlight and walking can sometimes make you feel much better because it releases endorphins. And, you know, sometimes that can be good. But it's sort of like this razor thin knife edge."

## Conclusion

In conclusion, individuals with long COVID report significant perceived health limitations, with physical and mental health ratings around one standard deviation lower than U.S. adult norms. Many noted low levels of PA, with nearly all reporting less PA than prior to spring 2020. Most interviewees reported negative physical and mental health effects of PA, including post-exertional malaise, worsening of specific long COVID symptoms during or following PA, a lack of control over PA level and abilities, a forced inactive lifestyle, and the loss of former PA abilities. However, some noted improvements in physical and mental health with PA, including improved strength and energy, a sense of accomplishment, and hope related to improvements in symptoms and PA abilities.

This study has limitations. Participants self-reported long COVID status, and all measures of health were self-reported and observational. Recruitment for the initial survey was primarily through mass email and long COVID-specific social media sites and those not engaged in these communities may have been excluded from the current study. The sample was relatively homogenous, consisting largely of individuals identifying as female, white, and not Hispanic or Latino, though the study did have greater representation of males than reported in other long COVID and PA studies [23,24]. Participant recruitment was limited to adults with long COVID based in the United States, and results cannot be extrapolated to individuals in other countries and children experiencing long COVID. Those completing the interviews may have differed in significant ways from those who did not participate, and it is possible that those who participated were less severely affected, having adequate functional abilities to engage in interviews. Those with strong opinions about long COVID or motivation for advocacy may have also been more likely to participate in the interview. Finally, measurement of physical activity did not involve a standardized questionnaire or objective assessment of physical activity such as through accelerometry, which may impact the accuracy of physical activity self-report.

This study provides insight into the varied perceived consequences of PA among individuals with long COVID, emphasizing the need to consider the lived experience in making PA recommendations. Future research, particularly analyses considering the lived experience, may help to further clarify the association between PA and perceived health among individuals with long COVID. Intervention studies assessing the impact of PA may consider following participants longitudinally, to allow assessment of both acute and chronic effects of PA on health among individuals with long COVID. Understanding the lived experience of PA in the context of long COVID will inform recommendations of healthcare providers and exercise professionals, influencing quality of life and health outcomes.

## Supporting information

**S1 Table.** Code descriptions.
(DOCX)

## Acknowledgments

The authors have no acknowledgements.

## Author contributions

**Conceptualization:** Zoe Sirotiak, Emily B.K. Thomas, Angelique G. Brellenthin.

**Data curation:** Zoe Sirotiak.

**Formal analysis:** Zoe Sirotiak, Arie M. Sirotiak, Kate A. Nettleton.

**Investigation:** Zoe Sirotiak, Arie M. Sirotiak, Kate A. Nettleton, Angelique G. Brellenthin.

**Methodology:** Zoe Sirotiak, Ann M. Oberhauser, Angelique G. Brellenthin.

**Project administration:** Angelique G. Brellenthin.

**Supervision:** Angelique G. Brellenthin.

**Writing – original draft:** Zoe Sirotiak.

**Writing – review & editing:** Zoe Sirotiak, Ann M. Oberhauser, Arie M. Sirotiak, Kate A. Nettleton, Duck-chul Lee, Emily B.K. Thomas, Angelique G. Brellenthin.

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
