## [Decision Letter · Decision Letter 0]

6 Apr 2026

PONE-D-25-63871Exploring the perceived impact of physical activity on physical and mental health among individuals with long COVID: A qualitative interview inquiryPLOS One

Dear Dr. Sirotiak,

Thank you for submitting your manuscript to PLOS ONE. After careful consideration, we feel that it has merit but does not fully meet PLOS ONE’s publication criteria as it currently stands. Therefore, we invite you to submit a revised version of the manuscript that addresses the points raised during the review process.

We look forward to receiving your revised manuscript.

Kind regards,

Wanli Zang, Ph.D.

Guest Editor

PLOS One

Reviewers' comments:

Reviewer's Responses to Questions

**Comments to the Author**

1. Is the manuscript technically sound, and do the data support the conclusions?

Reviewer #1: Yes

Reviewer #2: Yes

2. Has the statistical analysis been performed appropriately and rigorously? 

Reviewer #1: N/A

Reviewer #2: Yes

3. Have the authors made all data underlying the findings in their manuscript fully available?

Reviewer #1: No

Reviewer #2: Yes

4. Is the manuscript presented in an intelligible fashion and written in standard English?

Reviewer #1: Yes

Reviewer #2: No

5. Review Comments to the Author

Reviewer #1: Clinically relevant qualitative interview inquiry exploring the perceived impact of physical activity on physical and mental health among individuals with long COVID. Clinically relevant qualitative interview inquiry exploring the perceived impact of physical activity on physical and mental health among individuals with long COVID.

Reviewer #2: Dear Editor and authors,

Thank you very much for granting me the opportunity to review this manuscript entitled “Exploring the perceived impact of physical activity on physical and mental health among individuals with long COVID: A qualitative interview inquiry”. The study uses a qualitative interview approach to explore the perceived effects of physical activity on the physical and mental health of people with long COVID, and the topic has important practical significance. However, the manuscript still has some shortcomings. Below I will list the issues point by point and provide specific suggestions for revision based on the content of the paper.

1.Lack of objectivity in physical activity measurement

Basis: In the Methods section (lines 136–137), a single question was used for participants to self‑report “the number of minutes per week they typically engage in moderate or vigorous aerobic physical activity”, with rather coarse categories (0 minutes, 1–<150 minutes, 150–<300 minutes, ≥300 minutes). Is there any reference to support this classification? Compared with recognised physical activity assessment tools (e.g., the International Physical Activity Questionnaire, IPAQ), this approach lacks detail and validation.

Suggestion: Clearly state in the limitations section that this study did not use objective measurement devices (e.g., accelerometers) or a standardised questionnaire to assess physical activity, and that self‑report may be subject to recall bias and social desirability bias.

2.Incomplete description of ethics review details

Basis: In the Methods section (lines 104–106), it is stated that “the Iowa State University IRB deemed this project exempt and did not require formal written or verbal informed consent”, yet participants still read an information sheet and indicated consent by clicking “forward” on the screen. Whether such “click‑to‑consent” is ethically appropriate for a qualitative interview study is open to question.

Suggestion: Add to the ethics statement an explanation of why the IRB considered the project exempt (e.g., minimal risk, anonymised data handling). Also clearly state that, although formal written consent was not required by the IRB, the research team still provided participants with complete study information and obtained their clear informed consent.

3.Insufficient description of the interview guide development process

Basis: The Methods section (lines 127–129) mentions that the interview guide was developed by a physical therapist (ZS) with input from a qualitative methods expert (AO), a physical activity and health expert (AB), and a clinical psychologist (ET). However, it does not indicate whether the guide was pilot‑tested or whether revisions were made based on pilot results.

Suggestion: Add details about the development of the interview guide: Was a pilot interview conducted? If yes, what was the sample size of the pilot? What modifications were made to the guide after the pilot?

4.Inconsistency between the abstract and the main text

Basis: The abstract (lines 52–53) reports that “most participants (64.7%) reported worsening of long COVID symptoms with PA, while 14.7% reported improvement”. Table 2 shows 22 participants (64.7%) reporting worsening and 5 (14.7%) reporting improvement, so these numbers are consistent. The Discussion (line 199) also mentions “64.7% reported worsened health, 14.7% reported improvement”, and line 207‑208 mentions “20.6% reported no effect”. The sum of these percentages is 100%, which is consistent. However, when citing Wright et al. (2022) on lines 207‑208, the statement “28.7% reported no effect” is given without a direct comparison.

Suggestion: Add a direct comparison in the Discussion: the “no effect” proportion in this study was 20.6%, which is lower than the 28.7% reported by Wright et al. Possible reasons (e.g., sample differences, measurement differences) should be discussed.

5.Please use the terms “worsened / improved / no effect” consistently throughout the manuscript, avoiding different expressions for the same meaning.

6.The language of the manuscript needs further checking in places, such as tense and repeated or redundant expressions.

7.In “post‑acute sequalae of COVID‑19”, the word “sequalae” should be “sequelae”. Is this a spelling error? Please check the whole manuscript.

8.Lines 66–68: “Long COVID … affects up to 20% of those affected by COVID‑19 and over 65 million worldwide [1,2]”. Reference [1] was published in 2023, but the cited data may have been updated (e.g., more recent estimates from WHO or other sources). Please consider citing the most up‑to‑date data and literature.

9.The discussion of the “external control of PA” theme (lines 277–298) lacks theoretical support.

Issue: This theme relates to a perceived external control over physical activity, but no relevant psychological theories (e.g., Self‑Determination Theory, perceived control theory) are cited to explain this phenomenon.

Suggestion: Introduce the concept of “autonomy” from Self‑Determination Theory to explain why a sense of external control can be detrimental to the mental health of individuals with long COVID.

Thank you again for the opportunity to review this manuscript. I hope these comments are helpful for improving the paper.

6. PLOS authors have the option to publish the peer review history of their article (what does this mean?). If published, this will include your full peer review and any attached files.

Reviewer #1: No

Reviewer #2: **Yes:** Moran Lyu

---

## [Author Response · Author response to Decision Letter 1]

23 Apr 2026

Dear Dr. Zang,

We are submitting our revised manuscript “Exploring the perceived impact of physical activity on physical and mental health among individuals with long COVID: A qualitative interview inquiry” for reconsideration for publication in PLOS One. We have responded to each reviewer comment below. The authors thank the editor and reviewers for the careful review and thoughtful comments.

The resubmitted manuscript meets PLOS ONE’s style requirements as detailed at the provided links.

2. a) If there are ethical or legal restrictions on sharing a de-identified data set, please explain them in detail (e.g., data contain potentially identifying or sensitive patient information, data are owned by a third-party organization, etc.) and who has imposed them (e.g., a Research Ethics Committee or Institutional Review Board, etc.). Please also provide contact information for a data access committee, ethics committee, or other institutional body to which data requests may be sent. b) If there are no restrictions, please upload the minimal anonymized data set necessary to replicate your study findings to a stable, public repository and provide us with the relevant URLs, DOIs, or accession numbers. Please see http://www.bmj.com/content/340/bmj.c181.long for guidelines on how to de-identify and prepare clinical data for publication. For a list of recommended repositories, please see https://journals.plos.org/plosone/s/recommended-repositories. You also have the option of uploading the data as Supporting Information files, but we would recommend depositing data directly to a data repository if possible. Please update your Data Availability statement in the submission form accordingly.

The Data Availability statement has been uploaded to read:

“The data supporting the findings of this study include an interview guide and codebook. These materials are publicly available on the Open Science Framework at https://doi.org/10.17605/OSF.IO/6ZW4H. Full interview transcripts are not shared due to ethical and confidentiality considerations as approved by the Iowa State University Institutional Review Board. The Iowa State University Institutional Review Board may be contacted at irb@iastate.edu or 515-294-4566.”

A caption for the Supporting Information (S1 Table) has been added to the end of the manuscript.

Reviewer #1: Clinically relevant qualitative interview inquiry exploring the perceived impact of physical activity on physical and mental health among individuals with long COVID. Clinically relevant qualitative interview inquiry exploring the perceived impact of physical activity on physical and mental health among individuals with long COVID.

Thank you for your review.

Reviewer #2: Thank you very much for granting me the opportunity to review this manuscript entitled “Exploring the perceived impact of physical activity on physical and mental health among individuals with long COVID: A qualitative interview inquiry”. The study uses a qualitative interview approach to explore the perceived effects of physical activity on the physical and mental health of people with long COVID, and the topic has important practical significance. However, the manuscript still has some shortcomings. Below I will list the issues point by point and provide specific suggestions for revision based on the content of the paper.

Thank you for your detailed review of our manuscript.

1. Lack of objectivity in physical activity measurement

Basis: In the Methods section (lines 136–137), a single question was used for participants to self report “the number of minutes per week they typically engage in moderate or vigorous aerobic physical activity”, with rather coarse categories (0 minutes, 1–<150 minutes, 150–<300 minutes, ≥300 minutes). Is there any reference to support this classification? Compared with recognised physical activity assessment tools (e.g., the International Physical Activity Questionnaire, IPAQ), this approach lacks detail and validation.

Thank you for this comment. The authors agree that the lack of objective physical

activity measurement is a limitation of the current study. To minimize participant burden on the accompanying online survey, the researchers chose a single multiple-option survey item based on the established aerobic physical activity categories according to the 2018 Physical Activity Guidelines for Americans (0 min = inactive; 1 to <150 min = insufficiently active; 150 to <300 min = active; ≥300 min = highly active). This explanation has been added to the Methods section:

“Physical activity level. Participants selected whether they typically engaged in 0 minutes, 1 to <150 minutes, 150 minutes to <300, or ≥300 minutes or more of moderate or vigorous aerobic physical activity per week. These response options were selected based on established aerobic physical activity categories according to the 2018 Physical Activity Guidelines for Americans (0 min = inactive; 1 to <150 min = insufficiently active; 150 to <300 min = active; ≥300 min = highly active) [28].” (page 6, lines 140-143)

Suggestion: Clearly state in the limitations section that this study did not use objective measurement devices (e.g., accelerometers) or a standardised questionnaire to assess physical activity, and that self report may be subject to recall bias and social desirability bias.

Thank you for this helpful suggestion. We have now added this consideration to the

limitations section of our discussion:

“Finally, measurement of physical activity did not involve a standardized questionnaire or objective assessment of physical activity such as through accelerometry, which may impact the accuracy of physical activity self-report.” (page 17, lines 388-390)

2. Incomplete description of ethics review details. Basis: In the Methods section (lines 104–106), it is stated that “the Iowa State University IRB deemed this project exempt and did not require formal written or verbal informed consent”, yet participants still read an information sheet and indicated consent by clicking “forward” on the screen. Whether such “click to consent” is ethically appropriate for a qualitative interview study is open to question. Suggestion: Add to the ethics statement an explanation of why the IRB considered the project exempt (e.g., minimal risk, anonymised data handling). Also clearly state that, although formal written consent was not required by the IRB, the research team still provided participants with complete study information and obtained their clear informed consent.

Thank you for this comment. We have expanded on the nature of our study to further

contextualize the exemption granted by our Institutional Review Board. The Iowa State

University granted an exemption under Exemption 2 of the 2018 Common Rule for surveys, interviews, educational tests, and observation. With our project only involving surveys and interviews, the IRB conducted a “limited IRB review” and determined that our project had adequate privacy and confidentiality practices in place. The consent information sheet offered to participants prior to participation clearly described study procedures, with participants providing clear consent by clicking forward on the computer screen. This information is now included in the Methods section:

“The project was approved by the Iowa State University Institutional Review Board (IRB; IRB ID: 23-158). The Iowa State University IRB did not require formal written or verbal consent as this project was deemed exempt under Exemption 2 (surveys, interviews, educational tests, observation) of the 2018 Common Rule. This research project aligned with Exemption 2 in that it involved only survey and interview procedures, and the IRB conducted a “limited IRB review,” determining that adequate privacy and confidentiality practices were in place. However, participants read through a consent information sheet that contained complete study information. If they chose to participate, participants indicated their clear informed consent by clicking forward on the screen. Participants also gave verbal consent for recording of their interview prior to beginning the interview. Participants had the option to skip any question that they did not wish to answer during both the survey and the interview. Potentially identifying information in the transcripts was generalized prior to coding.” (pages 4-5, lines 100-111)

3. Insufficient description of the interview guide development process Basis: The Methods section (lines 127–129) mentions that the interview guide was developed by a physical therapist (ZS) with input from a qualitative methods expert (AO), a physical activity and health expert (AB), and a clinical psychologist (ET). However, it does not indicate whether the guide was pilot tested or whether revisions were made based on pilot results. Suggestion: Add details about the development of the interview guide: Was a pilot interview conducted? If yes, what was the sample size of the pilot? What modifications were made to the guide after the pilot?

Thank you for this comment. As we were performing semi-structured interviews that inherently varied regarding focus and specific questions based on participant responses, we did not perform pilot testing. The interview guide was developed with input from individuals with varying areas of expertise, allowing diverse perspectives to be incorporated into the interview guide. The number of eligible and interested participants as well as funding was also limited in the present study, limiting our ability to put resources towards pilot interviews. We have noted this reason in the Methods:

“The interview guide was developed by a physical therapist (ZS), with input from a qualitative methods expert (AO), a PA and health expert (AB), and a clinical psychologist (ET). Due to the semi-structured nature of the interview and limited available participants, pilot testing was not conducted, with interviews being flexible in response to participant experiences shared.” (pages 5-6, lines 128-131)

4. Inconsistency between the abstract and the main text

Basis: The abstract (lines 52–53) reports that “most participants (64.7%) reported worsening of long COVID symptoms with PA, while 14.7% reported improvement”. Table 2 shows 22 participants (64.7%) reporting worsening and 5 (14.7%) reporting improvement, so these numbers are consistent. The Discussion (line 199) also mentions “64.7% reported worsened health, 14.7% reported improvement”, and line 207 208 mentions “20.6% reported no effect”. The sum of these percentages is 100%, which is consistent. However, when citing Wright et al. (2022) on lines 207 208, the statement “28.7% reported no effect” is given without a direct comparison. Suggestion: Add a direct comparison in the Discussion: the “no effect” proportion in this study was 20.6%, which is lower than the 28.7% reported by Wright et al. Possible reasons (e.g., sample differences, measurement differences) should be discussed.

Thank you for this suggestion. We have adjusted this sentence to clarify that the

percent of participants noting no effect of PA on long COVID symptoms in our study (20.6%) was lower than the 28.7% reported by Wright et al. (2022). We have also added possible explanations for this discrepancy, including differences in sample size (34 in our study, 477 in Wright et al. 2022) and the assessment of impact at different physical activity intensities in Wright et al. (2022) compared to non-specific physical activity in the current study. These additions are present in the Discussion section:

“The percentage of participants noting no effect of PA on long COVID symptoms in the present study (20.6%) was lower than the 28.7% reported by Wright et al. (2022). Differences in sample size and consideration of physical activity intensities by Wright et al. (2022) compared to non-specific physical activity in the current study may contribute to these differences noted.” (pages 9-10, lines 220-224)

5. Please use the terms “worsened / improved / no effect” consistently throughout the manuscript, avoiding different expressions for the same meaning.

Thank you for this suggestion. We have ensured that these terms are utilized effectively throughout the manuscript to allow more consistent wording and clearer interpretations.

6. The language of the manuscript needs further checking in places, such as tense and repeated or redundant expressions.

The complete manuscript has been checked for accuracy of spelling, grammar and redundancies.

7. In “post acute sequalae of COVID 19”, the word “sequalae” should be “sequelae”. Is this a spelling error? Please check the whole manuscript.

Thank you for pointing this out. This spelling mistake has been corrected. Additionally, the rest of the manuscript has been reviewed for accuracy.

8. Lines 66–68: “Long COVID … affects up to 20% of those affected by COVID 19 and over 65 million worldwide [1,2]”. Reference [1] was published in 2023, but the cited data may have been updated (e.g., more recent estimates from WHO or other sources). Please consider citing the most up to date data and literature.

Thank you for this suggestion. We have updated these estimates with figures provided by more recent literature. We utilized estimates based on a multi-year study utilizing a network of electronic health records (adult estimate: 10-26%) and a meta-analysis (estimate: 36%) assessing long COVID prevalence across several years. We have updated this figure in our Introduction:

“Long COVID, or the experience of new or worsened chronic symptoms following COVID-19 infection, affects between 10-36% of adults following SARS CoV-2 infection [1–4].” (page 3, lines 62-63)

We have also added more recent references to the Introduction, as the literature surrounding long COVID has continued to grow.

9. The discussion of the “external control of PA” theme (lines 277–298) lacks theoretical support. Issue: This theme relates to a perceived external control over physical activity, but no relevant psychological theories (e.g., Self Determination Theory, perceived control theory) are cited to explain this phenomenon. Suggestion: Introduce the concept of “autonomy” from Self Determination Theory to explain why a sense of external control can be detrimental to the mental health of individuals with long COVID.

Thank you for this helpful suggestion. We have now incorporated the concept of autonomy, as well as competence from self-determination theory, noting that participants in our study often noted challenges to their feelings of autonomy and competence regarding physical activity, aligning with other roles and responsibilities affected as noted within the long COVID literature. This added paragraph to the Discussion section reads:

“Self-determination theory, which posits that humans have three innate psychological needs [46], may be a framework through which the external control of PA noted may be understood. These three proposed needs of self-determination theory include competence, autonomy, and relatedness, with motivation and quality of life significantly impacted if individuals are unable to meet them [46,47]. Individuals with long COVID have noted loss of autonomy related to symptoms, resulting in wide-ranging impacts across social and professional roles [48]. PA experiences among participants in the present study seemed to align with published literature, suggesting that autonomy may be limited by poor personal experiences with PA, distrust of healthcare provider PA recommendations, and feeli

---

## [Decision Letter · Decision Letter 1]

10 May 2026

Exploring the perceived impact of physical activity on physical and mental health among individuals with long COVID: A qualitative interview inquiry

PONE-D-25-63871R1

Dear Dr. Sirotiak,

We’re pleased to inform you that your manuscript has been judged scientifically suitable for publication and will be formally accepted for publication once it meets all outstanding technical requirements.

Kind regards,

Wanli Zang, Ph.D.

Guest Editor

PLOS One

Additional Editor Comments (optional):

Reviewers' comments:

Reviewer's Responses to Questions

**Comments to the Author**

1. If the authors have adequately addressed your comments raised in a previous round of review and you feel that this manuscript is now acceptable for publication, you may indicate that here to bypass the “Comments to the Author” section, enter your conflict of interest statement in the “Confidential to Editor” section, and submit your "Accept" recommendation.

Reviewer #2: All comments have been addressed

2. Is the manuscript technically sound, and do the data support the conclusions?

Reviewer #2: Yes

3. Has the statistical analysis been performed appropriately and rigorously? 

Reviewer #2: Yes

4. Have the authors made all data underlying the findings in their manuscript fully available?

Reviewer #2: Yes

5. Is the manuscript presented in an intelligible fashion and written in standard English?

Reviewer #2: Yes

6. Review Comments to the Author

Reviewer #2: This revision has significantly improved the quality of the manuscript, and all the previously raised issues have been well addressed. I have no further suggestions for revision, as the manuscript in its current form is of high quality.

7. PLOS authors have the option to publish the peer review history of their article (what does this mean?). If published, this will include your full peer review and any attached files.

Reviewer #2: **Yes:** Moran Lyu

---

## [Editor Report · Acceptance letter]

PONE-D-25-63871R1

PLOS One

Dear Dr. Sirotiak,

I'm pleased to inform you that your manuscript has been deemed suitable for publication in PLOS One. Congratulations! Your manuscript is now being handed over to our production team.

Kind regards,

on behalf of

Dr. Wanli Zang

Guest Editor

PLOS One